# Vapur: A Search Engine to Find Related Protein - Compound Pairs in COVID-19 Literature

**Abdullatif Köksal**
Computer Engineering,
Boğaziçi University, Turkey
abdullatif.koksal@boun.edu.tr

**Hilal Dönmez**
Computer Engineering,
Boğaziçi University, Turkey
hilal.donmez@boun.edu.tr

**Rıza Özçelik**
Computer Engineering,
Boğaziçi University, Turkey
riza.ozcelik@boun.edu.tr

**Elif Ozkirimli**
Chemical Engineering,
Bogazici University, Turkey
Data and Analytics Chapter,
F. Hoffmann-La Roche AG, Switzerland
elif.ozkirimli@roche.com

**Arzucan Özgür**
Computer Engineering,
Boğaziçi University, Turkey
arzucan.ozgur@boun.edu.tr

## Abstract

Coronavirus Disease of 2019 (COVID-19) created dire consequences globally and triggered an intense scientific effort from different domains. The resulting publications created a huge text collection in which finding the studies related to a biomolecule of interest is challenging for general purpose search engines because the publications are rich in domain specific terminology. Here, we present Vapur: an online COVID-19 search engine specifically designed to find related protein - chemical pairs. Vapur is empowered with a relation-oriented inverted index that is able to retrieve and group studies for a query biomolecule with respect to its related entities. The inverted index of Vapur is automatically created with a BioNLP pipeline and integrated with an online user interface. The online interface is designed for the smooth traversal of the current literature by domain researchers and is publicly available at https://tabilab.cmpe.boun.edu.tr/vapur/.

## 1 Introduction

Coronavirus Disease of 2019 (COVID-19) outbreak had severe impacts on human health all around the world since December 2019, but also triggered an exceptional amount of scientific work. As of September 2020, PubMed recorded over 45K articles (Chen et al., 2020) related to COVID-19 since its inception, including works on diagnosis (Dorche et al., 2020), drug repurposing (Gao et al., 2020), and text-mining (Tarasova et al., 2020). As the body of literature keeps growing in the form of unstructured text, it also becomes more and more challenging for researchers to find the relevant information they need. Furthermore, the publications include domain-specific named entities and relations that challenge the general-purpose search engines. Therefore, it is of critical importance to build a search engine that can find relevant documents in this terminology-rich and domain-specific literature.

Biomedical named entity recognition (NER) and relation extraction can be utilized to semantically structure publications around the biochemically related entities. When named entities and their relations are extracted, a document can be expressed as a set of triplets of the form $(Entity1, Entity2, Relation)$. This formulation can be converted to an inverted index from the related entities to the publications in order to enable retrieving relevant documents to a query by entity and relation matching. If the same entities are referenced with different words (e.g. ACE2, Angiotensin-converting enzyme 2, Q9BYF1), named entity normalization can be used to identify different mentions of the same entity. Enhanced with named entity normalization, the inverted index can retrieve the documents that contain biochemical relations of a free-text query, as grouped by the related biomolecules.

In this work, we present Vapur, an online search engine to find related protein - compound pairs in the COVID-19 anthology. Vapur is empowered with a biochemical relation-based inverted index that is created through named entity recognition, named entity normalization, and relation extraction on CORD-19 abstracts (Wang et al., 2020a). Thanks to the underlying biochemical domain-specific tools and relation extraction model, Vapur identifies biochemically related entities to a free-text query and retrieves the publications that mention the relation. The design of Vapur offers a novel approach to explore COVID-19 literature with a fo-

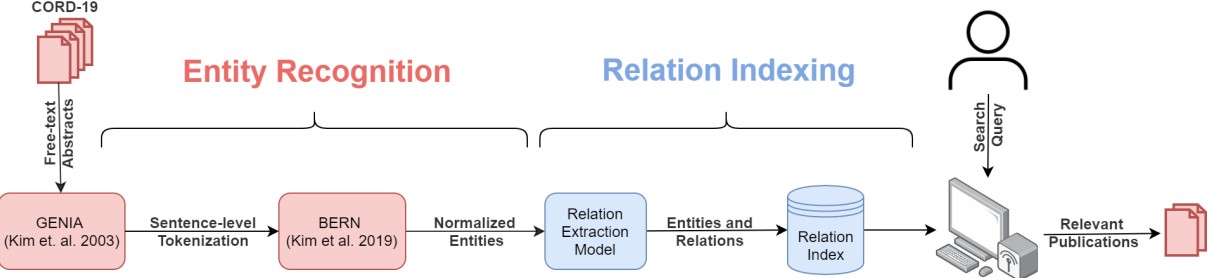

Figure 1: The workflow of the pipeline behind Vapur. We first split CORD-19 abstracts to sentences and use BERN to detect and normalize the entities in the text. We then identify the biochemical relations with the relation extraction model that we trained and reform the output as an inverted index of relations. Vapur leverages this inverted index to retrieve relevant publications to the query as categorized by related entities.

cus on biomolecules and their relations. We present Vapur at `https://tabilab.cmpe.boun.edu.tr/vapur/` and publicly share the code and models at `https://github.com/boun-tabi/vapur`.

## 2 Related Work

Keeping up to date with the growing body of biomedical literature is a big challenge and almost a kind of magic in scientific research. Efforts such as ChemProt (Krallinger et al., 2017) aim to promote research in this direction by presenting PubMed abstracts in which proteins, chemicals, and their relations are manually labeled. Liu et al., 2018 and Lim and Kang, 2018 developed sequential models for chemical and protein relation extraction on ChemProt, while Peng et al., 2018 used an ensemble of deep models with SVM. However, all these models depend on the named entities to be recognized beforehand.

Named entity recognition and normalization are widely studied topics in biomedical text mining to extract and link entities. Conditional Random Fields (CRF) was a popular approach in early biomedical NER studies (Leaman et al., 2015; Wei et al., 2015, 2018) and with the rise of deep learning, sequential models were also integrated (Sachan et al., 2018). Recently, transformers-based NER models attracted more attention, including BERN (Kim et al., 2019), which is a state-of-the-art biomedical named entity recognition and normalization tool that uses BioBERT (Lee et al., 2020b) to identify and normalize the entities in a sentence. BERN is adopted by recent general-domain biomedical text mining studies (Jang et al., 2019) as well as a model to answer COVID-19 related questions based on CORD-19 (Lee et al., 2020a).

CORD-19 (Wang et al., 2020a) is a data set comprising scientific publications related to COVID-19 and it has provided a valuable resource for text mining studies (Su et al., 2020; Esteva et al., 2020; Wang et al., 2020b), including COVID-19 specific search engine development such as SemViz (Tu et al., 2020) and KD-COVID[1].

SemViz aims to present subtle relations between entities by semantic visualization of a CORD-19 based knowledge-graph. KD-COVID, on the other hand, identifies the most similar sentence in CORD-19 to a query and retrieves the corresponding publication. KD-COVID also provides links to large protein, disease, and protein-disease relation databases. Vapur is different from these approaches, since it is based on an inverted index from biochemically related entities to publications. While most other systems depend on existing knowledge bases to provide information about related entities, Vapur uses NLP techniques to automatically extract the most up-to-date information from the scientific literature.

## 3 Vapur

### 3.1 System Description

Vapur is an online search engine with a focus on finding related proteins and chemicals in the COVID-19 literature. Vapur is able to retrieve relevant documents to a query as categorized by the biochemically related entities thanks to its relation-oriented inverted index. In order to obtain the index, Vapur first identifies and normalizes the named entities in CORD-19 abstracts using a pre-trained model, BERN (Kim et al., 2019). Afterward, Vapur determines if the entity pairs in the same sentence are related to each other by the binary relation ex-

---

[1] `http://kdcovid.nl/about.html`

traction model we trained on the ChemProt data set (Krallinger et al., 2017). The result of the relation extraction model is a list of related entities for each abstract. This list is then used to construct the inverted index that represents biochemical relations as entity pairs and maps each relation to the documents in which the relation was mentioned. Vapur is publicly available at `https://tabilab.cmpe.boun.edu.tr/vapur/` and Figure 1 illustrates its workflow.

Vapur represents the entities as equivalence classes learned from the named entity recognition and normalization steps. For instance, "Interleukin-1b" is the member of the class {interleukin-1b, IL-1beta, IL1B, HGNC:5992, . . . , BERN:323737602} in the index and Vapur retrieves related entities and relevant documents with a mention "Interleukin-1b" even if the query is "IL1B". The equivalence classes cover a wide range of mention types from free-text to chemical IDs and enable a flexible search experience for the search terms.

Vapur represents each entity mention as a string and adopts a 3-gram based matching algorithm to search the queried entity in its index. Given a query, Vapur first creates a multi-set of all 3-grams of the query and computes the similarity of this set to all 3-gram multi-sets of the mentions in the index, which are pre-computed. Vapur measures the similarity of the query to a mention by generalized Jaccard similarity:

$$J(Q, M) = \frac{\sum_i min(Q_i, M_i)}{\sum_i max(Q_i, M_i)}$$

where $Q$ and $M$ are query and mention vectors that store the count of each 3-gram. Generalized Jaccard similarity makes Vapur more robust in retrieving results for typos in a query.

When the related entities and their abstracts are retrieved, Vapur ranks the related entities by the number of times they are co-mentioned with the queried entity. Each entity is displayed with its most frequent mention alongside the links to the papers.

We also interpret the relation-oriented index of Vapur as a graph to find similar entities to the query that cannot be trivially inferred from the inverted index. We construct a graph from the index such that the nodes represent biomolecules and the edges denote biochemical relations. The resulting graph satisfies the bipartiteness property, since we identify the relations only between the proteins and

| Graph Property | Statistic |
| --- | --- |
| # Nodes (Entities) | 12384 |
| # Chemical Nodes | 5018 |
| # Protein Nodes | 7366 |
| # Edges (Relations) | 17657 |
| # Connected Components | 807 |
| # Nodes in the Largest Component | 10194 |
| # Edges in the Largest Component | 16226 |
| # Diameter of the Largest Component | 18 |

Table 1: Summary statistics for the bipartite graph of the inverted index. We observe that number of chemicals and proteins are comparable to each other and the number of relations is close to the number of nodes, indicating the sparsity. In addition, the graph is formed of 807 connected components but the largest one is significantly larger than the others.

chemicals during binary relation extraction. Table 1 demonstrates the summary statistics of this bipartite graph computed via networkx (Hagberg et al., 2008).

We leverage the bipartite graph structure to identify similar biomolecules of the same type. We use SimRank (Jeh and Widom, 2002) to compute pairwise node similarity and list the five entities most similar to the query, in addition to the search results. Our goal is to allow researchers to explore information in the literature not only for the query, but also for entities similar to the query.

### 3.2 Use Case Scenario

We consulted with domain experts to build scenarios in which Vapur can help researchers. We present a use case that highlights how the relation-based index of Vapur and the similar entity prediction can extend the scope of a research. We illustrate the scenario in Figure 2.

**Relation between Favipiravir and RdRp:** A medicinal chemist working on COVID-19 drug development is interested in "Favipiravir", one of the drugs clinically used in COVID-19 therapy. She searches Favipiravir on Vapur and the results highlight RNA-dependent RNA polymerase (RdRp) enzyme as the most frequently co-mentioned protein that is identified by Vapur as being related to Favipiravir. Vapur displays sentences from the publications stating that Favipiravir inhibits RdRp and also proposes "Examorelin" as a contextually similar molecule to Favipiravir. The researcher decides to extend her research to Examorelin, another drug

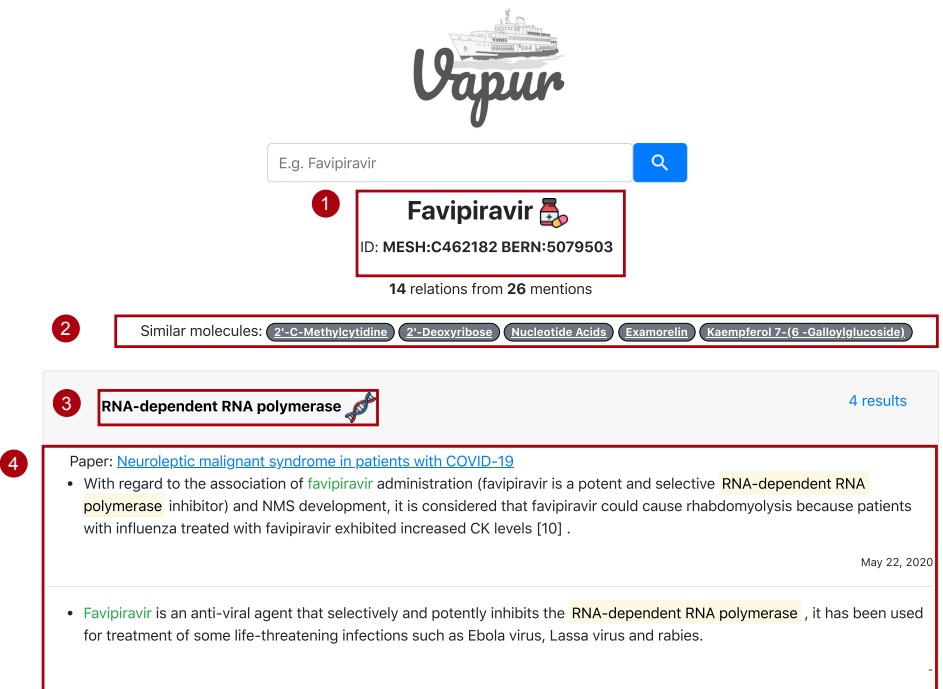

Figure 2: A search scenario on Vapur. When *Favipiravir* is entered as the query, Vapur shows the query and its unique ids in **1**. It lists contextually similar biomolecules to Favipiravir in **2**. It displays related biomolecules found by the relation extraction model in **3**. For example, the first result is RNA-dependent RNA polymerase for Favipiravir query. Finally, it shows sentences in related papers to provide information about the relations as in **4**.

known to bind RdRp.

## 4 Methods

### 4.1 Data sets

**CORD-19**  We create a relation-oriented search engine for CORD-19 abstracts. CORD-19 is a regularly updated data set that contains studies related to COVID-19 and SARS-CoV-2. We use the CORD-19 snapshot of August 23, 2020, that contains ≈ 233K documents for which ≈ 143K unique abstracts are provided. Vapur indexes these abstracts, but is also able to return the linked full-paper.

**ChemProt**  We trained a binary relation extraction model using the ChemProt data set (Krallinger et al., 2017), a curation of PubMed abstracts whose entities and relations are manually annotated. ChemProt consists of 2432 abstract, 62,147 entity, and 45,171 relation files. An abstract file contains a PubMed abstract ID and its text, whereas an entity file lists the chemicals and proteins in the linked abstract. Last, a relation file reports the relation type between entity pairs in the same sentence whose relation types can be inferred from the sentence.

The relations in ChemProt are categorized into 11 categories, where the first 10 categories (CPR:0 to CPR:9) denote various types of biochemical relations, while the last category (CPR:10) is reserved for "not relation" (i.e., the sentence explicitly states that there is no relation between the specified pair of entities). In this work, we focus on binary relation extraction, where we aim to determine whether a sentence states that there is a relation between the specified two entities or not. Therefore, we consider the entity pairs in the first 10 categories as positive samples. We treat the CPR:10 and the "Other" categories as the negative class, since the sentences for these two categories do not state that there is a relation between the corresponding pair of entities. We describe each relation type and report their instance counts in the ChemProt data set in Table 2. We also provide summary statistics for ChemProt in Table 3.

### 4.2 Named Entity Recognition and Normalization

We used BERN (Kim et al., 2019), a neural NER architecture with an integrated normalizer that can perform in a multi-type setting, to identify and normalize the entities in CORD-19 abstracts. BERN is an ensemble of existing tools and models (Wei

| Relation ID | Description | Train Count | Dev Count | Test Count | Binary Label |
|---|---|---|---|---|---|
| CPR:0 | *No description provided by ChemProt* | 1 | 2 | 0 | 1 |
| CPR:1 | Part of | 308 | 153 | 215 | 1 |
| CPR:2 | Regulator / Direct Regulator / Indirect regulator | 1652 | 780 | 1743 | 1 |
| CPR:3 | Upregulator / Activator/ Indirect upregulator | 777 | 552 | 667 | 1 |
| CPR:4 | Downregulator / Inhibitor / Indirect Downregulator | 2260 | 1103 | 1667 | 1 |
| CPR:5 | Agonist / Agonist activator / Agonist inhibitor | 173 | 116 | 198 | 1 |
| CPR:6 | Antagonist | 235 | 199 | 293 | 1 |
| CPR:7 | Modulator / Modulator Activator / Modulator inhibitor | 29 | 19 | 25 | 1 |
| CPR:8 | Cofactor | 34 | 2 | 25 | 1 |
| CPR:9 | Substrate / Product of / Substrate product of | 727 | 457 | 644 | 1 |
| CPR:10 | Not relation | 241 | 175 | 267 | 0 |
| *Other* | *No relation information between entities in this context* | 11664 | 7780 | 9987 | 0 |
| **Total** | | 18101 | 11338 | 15731 | |

Table 2: Chemical - Protein Relations (CPR) in ChemProt. The entity pairs are annotated with 11 different relation types in ChemProt and we refer to pairs whose relation information cannot be inferred from the context as *Other*. We report the instance count of each category and specify the label we assign to each category for binary relation extraction.

| Statistics | Train | Dev | Test |
|---|---|---|---|
| # documents | 1020 | 612 | 800 |
| # relations | 18046 | 11294 | 15712 |
| # entities | 25752 | 15567 | 20828 |
| # chemical mentions | 13017 | 8004 | 10810 |
| # unique chemical entities | 3710 | 2517 | 3442 |
| # protein mentions | 12735 | 7563 | 10018 |
| # unique protein entities | 4610 | 3018 | 3757 |
| # duplicate relations | 98 | 86 | 158 |
| # positive labels | 6143 | 3339 | 5459 |
| # negative labels | 11903 | 7955 | 10253 |

Table 3: ChemProt summary statistics. We split abstracts to sentences via GENIA (Kim et al., 2003) and compute the statistics accordingly and therefore some relations in ChemProt that are not in a sentence are filtered out. We refer to relations in the same sentence with the same protein, chemical and relation type as duplicate relations and consider them only once during training. We observe that the number of mentions is considerably higher than the number of unique entities for both proteins and chemicals, emphasizing the need for normalization. In addition, negative entity pairs are significantly more frequent than the positive ones, but their distribution across the training, development, and test folds is similar.

et al., 2015; Leaman et al., 2015; D'Souza and Ng, 2015; Wei et al., 2016, 2018; Lee et al., 2020b) and outputs a set of IDs for each recognized entity, given a sentence.

In this study, we first tokenized the abstracts with GENIA (Kim et al., 2003) and identified 1.23M sentences. We used BERN to extract named entities and their IDs in these sentences and recognized 1.58M entities in total, in which 171K are chemicals, 318K are proteins, and others are diseases and species. With normalization, we computed the number of unique chemicals and proteins as 20K and 70K, respectively.

### 4.3 Relation Extraction

We propose a BioBERT-based model to identify related entities in CORD-19 abstracts. To this end, we first preprocessed the sentences to explicitly encode the entities in the input and then finetuned BioBERT with the preprocessed sentences in ChemProt. We used the binary labels in Section 4.1 as outputs.

**Preprocessing**   We surrounded the named entities in the sentences with opening and closing tags to mark their location. We tagged chemicals with $<e1>$ and $</e1>$ and proteins with $<e2>$ and $</e2>$ to encode entity type and location during training. When a sentence had multiple chemical - protein pairs, we considered each pair separately

| Raw Sentence | EGFR inhibitors currently under investigation include the small molecules gefitinib and erlotinib. |
|---|---|
| **Preprocessed Form I** | <e2>EGFR</e2> inhibitors currently under investigation include the small molecules <e1>gefitinib</e1> and erlotinib. |
| **Preprocessed Form II** | <e2>EGFR</e2> inhibitors currently under investigation include the small molecules gefitinib and <e1>erlotinib</e1>. |

Table 4: An example of preprocessing. The raw sentence contains two chemicals (*gefitinib*, *erlotinib*) and one protein (EGFR), creating two protein - chemical pairs. Thus, we create two different forms of the sentence to encode each protein - chemical pair separately. We use <e1> and <e2> tags to enclose chemicals and proteins, respectively.

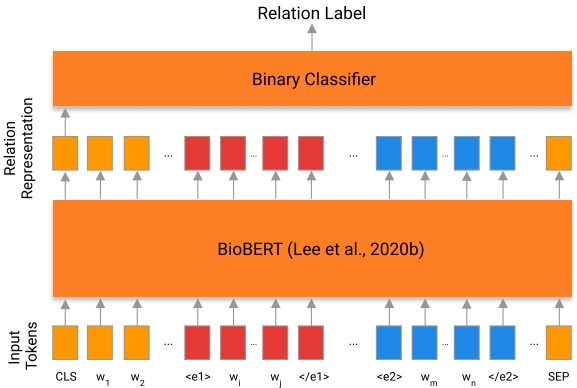

Figure 3: Relation Extraction Model. Our model is composed of a BioBERT and a binary classification layer. We tokenize the input sentences by BioBERT tokenizer and enclose the biochemical entities with special tags (<e1> and <e2>). BioBERT uses special tokens: *CLS* and *SEP* tags for fixed length-relation representation and sentence separation, respectively. We use BioBERT output of *CLS* token to train our binary classifier.

and created copies of the sentence with different tags. Table 4 provides an example of input and its two preprocessed forms.

**Model** We trained a binary model that decides if two entities in the same sentence are biochemically related. The goal of training a binary model instead of a multi-class one is to identify biochemical relations beyond the classes in ChemProt. By formulating the problem as binary classification, we can potentially identify the biochemically related molecules in CORD-19, whose biochemical relation type cannot be categorized under any of the ChemProt classes.

We used BioBERT with the addition of a single-layer binary log-softmax classifier and trained the classifier with the inputs and outputs from the previous step using the cross-entropy loss. BioBERT creates a fixed-length relation representation for the

starting tag *CLS* and we used this representation to vectorize the sentences. Figure 3 illustrates our relation extraction model.

**Experimental setup** We conducted a hyperparameter search with different optimizers, learning rates, and weight decays. We trained a model 10 times per parameter combination and selected the best one based on the F-score on the development set. We found that the AdamW optimizer with a 0.00003 learning rate and 0.1 weight decay yields the best results. We selected the best setting for relation extraction and computed mean precision, recall, and F1-score as well as the standard deviation on the ChemProt development (dev) and test sets.

## 5 Results and Discussion

In order to assess the performance of Vapur, we evaluated the performance of its components both as separate modules and together as an end-to-end pipeline. Since BERN, the named entity recognition and normalization module, has already been shown to be successful (Kim et al., 2019), we focus on automatic evaluation of the relation extraction model and expert evaluation of Vapur in this work.

### 5.1 Relation Extraction Model

We built the relation extraction model by fine-tuning a binary classifier on BioBERT and then computed precision, recall, and F1-Score on the dev and test sets of ChemProt. We also fine-tuned the same classifier using English BERT based model and report the results in Table 5.

Table 5 demonstrates that BioBERT obtained higher scores in terms of all three metrics on both folds, indicating that BioBERT is superior to BERT as a pre-trained language model for relation extraction on ChemProt. We relate this with the fact that BioBERT is trained with a more domain-related

| Fold | Model | Precision | Recall | F1-Score |
|------|-------|-----------|--------|----------|
| Dev | BERT | 0.718±0.027 | 0.737±0.026 | 0.727±0.010 |
| | **BioBERT** | **0.742±0.036** | **0.829±0.035** | **0.782±0.012** |
| Test | BERT | 0.759±0.023 | 0.710±0.026 | 0.733±0.007 |
| | **BioBERT** | **0.791±0.026** | **0.766±0.036** | **0.777±0.010** |

Table 5: BERT- and BioBERT-based fine-tuning results for binary relation extraction. We computed precision, recall, and F1 on the dev and test folds of ChemProt. The results show that BioBERT-based model outperforms BERT-based model in terms of all metrics on both folds.

| Relation ID | Test Set Accuracy |
|-------------|-------------------|
| CPR:1 | 0.661 ± 0.053 |
| CPR:2 | 0.667 ± 0.043 |
| CPR:3 | 0.855 ± 0.015 |
| CPR:4 | 0.874 ± 0.027 |
| CPR:5 | 0.850 ± 0.045 |
| CPR:6 | 0.856 ± 0.050 |
| CPR:7 | 0.890 ± 0.044 |
| CPR:8 | 0.550 ± 0.066 |
| CPR:9 | 0.617 ± 0.060 |
| CPR:10 | 0.828 ± 0.053 |
| *Other* | 0.892 ± 0.020 |

Table 6: Test performance of the relation extraction model by CPR label. We used the best model out of 10 models trained with different random seeds to predict test set pairs and report mean and standard deviation of the test accuracy by label.

text and BERT observed texts from a wider range.

We further investigated the performance of our relation extraction model by computing the test accuracy per CPR category and illustrate the results in Table 6. We observe that the relation extraction model achieved the highest accuracy in the *Other* category, indicating that the model can successfully identify whether the context is sufficient to deduce a relation between the entities or not. We relate the high performance for the *Other* category by using a contextual model for relation representation, BioBERT. Context-awareness enables Vapur to eliminate the documents that mention the queried entity in irrelevant contexts and to retrieve a document only if it contains relation information for the query.

## 5.2 Vapur

We evaluated Vapur from two different perspectives. We first analyzed 41 sample sentences in which Vapur identified a biochemical relation as a first step to discover the limitations of the complete pipeline. Then, we asked six biologists/chemists to use Vapur and rate its different aspects to demonstrate the success and usefulness of Vapur for future research.

Our inspection of 41 sample sentences indicated that most of the incorrect relation labels were due to incorrect entity assignment by BERN. In some cases, parts of the protein sequence such as *N-terminal*, *carboxyl terminal* or residue names such as *Asp238* are recognized as compounds. Table 7 illustrates sample sentences with incorrectly labeled entities. Other examples that were manually checked by a domain expert are presented in the Appendices.

In order to evaluate the real-life usefulness of Vapur, we asked six domain experts to use Vapur for five COVID-19 related queries (totalling up to 30) of their own. They each filled in a questionnaire where for each query they indicated (i) if each of the top three search results is related to the query, (ii) if similar molecules predicted by Vapur are in fact useful, and (iii) if the extracted sentences are useful. They also rated the ease of use of Vapur between 1 (very difficult) and 5 (very easy) and assessed its usefulness for future research on COVID-19.

The expert evaluations demonstrated that 27 out of 30 (90%) top search results and 76 out of 90[2] (84%) top three search results are biochemically related to the query, suggesting that Vapur successfully retrieves biochemically related entities to the query. During these experiments, Vapur retrieved at least one related document for each query. It returned only one result for two of the queries, which caused 4 of the 14 unsuccessful cases. These evaluations suggest that the inverted index of Vapur spans a comprehensive range of entities and contains sufficient number of documents to find a biochemically related result for each of the tested queries.

For each of the 30 queries of the domain experts, 5 similar molecule predictions are returned, resulting in a total of 150 molecule predictions. The questionnaire results indicated that 78 out of 150

---

[2]Note that for 30 queries there are a total of 30x3=90 top three search results.

| Sentence in CORD-19 | Incorrectly Labeled Entity |
|---|---|
| \<e1\>Alanine\</e1\> substitution of either Arg-76 or Tyr-94 in the N-terminal domain of \<e2\>IBV N protein\</e2\> led to a significant decrease in its RNA-binding activity and a total loss of the infectivity of the viral RNA to Vero cells. | **Alanine** |
| \<e2\>Rat microsomal aldehyde dehydrogenase\</e2\> (msALDH) has no amino-terminal signal sequence, but instead it has a characteristic hydrophobic domain at the \<e1\>carboxyl\</e1\> terminus (Miyauehi, K., R. | **carboxyl** |
| Also, the \<e2\>protease Factor Xa\</e2\>, a target of \<e1\>Ben\</e1\>-HCl abundantly expressed in infected cells, was able to cleave the recombinant and pseudoviral S protein into S1 and S2 subunits, and the cleavage was inhibited by Ben-HCl. | **Ben** |

Table 7: Sample sentences with incorrectly labeled entities. The entity types of *Alanine, carboxyl*, and *Ben* were incorrectly predicted as compounds by BERN during the named entity recognition step. Consequently, the relation extraction model incorrectly identified a biochemical relation.

(52%) similar molecule predictions were identified as correct by the domain experts. For 14 out of 30 queries, 4 or 5 of the predicted molecules were identified as correct suggesting that Vapur was able to identify similar molecules successfully for about half of the queries. On the other hand, for 9 out of 30 queries, none of the similar molecule predictions were identified as correct by the domain experts. We should note that the definition of "similarity" may be subjective. Here, Vapur extracts similar molecules if they are contextually similar to the query molecule. Therefore, the molecules may or may not be structurally similar.

The evaluation further showed that for 22 out of the 30 queries (73%) the extracted sentences are useful for research. Besides, the experts unanimously expressed that Vapur is very easy to use (5/5) and useful for future research.

Overall, both manual inspection and expert evaluations showed that Vapur can successfully find biochemically related proteins and chemicals in CORD-19 and can help future research on COVID-19.

## 6 Conclusion

We present Vapur, a search engine with an emphasis on identifying protein - chemical relations in the COVID-19 domain. To the best of our knowledge, Vapur is the first search engine that uses relation extraction to construct an inverted index of related biochemical entities beyond knowledge bases. Thanks to the relation extraction model, Vapur is able to categorize documents relevant to a query by biochemical entities related to it.

We evaluated the relation extraction model on ChemProt and observed that BioBERT-based model has the highest performance. Expert evaluation of the sample annotated sentences from CORD-19 showed that Vapur successfully finds relations between entities when the entities are identified and normalized accurately. Finally, six domain experts used Vapur with 5 different queries and expressed that Vapur is easy to use, retrieves results that are related to the queries, and it is useful for future research.

Although we focused on COVID-19 in this work, our approach can find application in any biomedical domain with a different choice of database to index. We believe that domain-specific scientific search engines will gain more interest in the future since COVID-19 is unlikely to be the last global health crisis. Consequently, we plan to improve Vapur by integrating full-papers, developing more robust named entity recognizers and normalizers, and indexing text databases on different domains. In order to help similar efforts, we make the underlying pipeline and Vapur available at https://github.com/boun-tabi/vapur and https://tabilab.cmpe.boun.edu.tr/vapur/, respectively.

## Acknowledgments

We would like to thank our domain experts, Asu Büşra Temizer, Eren Can Ekşi, Prof. Hacer Karataş, İrem Şenman, Prof. Nilgün Karalı, Serhat Beyaz, and Yağmur Ersoy for their detailed evaluation of Vapur.

TUBITAK-BIDEB 2211-A Scholarship Program (to A.K. and R.Ö.), and TUBA-GEBIP Award of the Turkish Science Academy (to A.Ö.) are gratefully acknowledged. E.O. is an employee of F. Hoffman - La Roche AG, Switzerland.

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

# 7 Appendices

| Sentence | Identified Chemical | Identified Protein | Relation Label | Chemical Label | Protein Label |
|---|---|---|---|---|---|
| The N-terminal domain of the coronavirus nucleocapsid (N) protein adopts a fold resembling a right hand with a flexible, positively charged b-hairpin and a hydrophobic palm. | N | coronavirus nucleocapsid (N) protein | 0 | 0 | 1 |
| Alanine substitution of either Arg-76 or Tyr-94 in the N-terminal domain of IBV N protein led to a significant decrease in its RNA-binding activity and a total loss of the infectivity of the viral RNA to Vero cells. | Alanine | IBV N protein | 0 | 0 | 1 |
| Rat microsomal aldehyde dehydrogenase (msALDH) has no amino-terminal signal sequence, but instead it has a characteristic hydrophobic domain at the carboxyl terminus (Miyauehi, K., R. | carboxyl | msALDH | 0 | 0 | 1 |
| The IMPDH inhibitor merimepodib (MMPD) is an investigational antiviral drug that acts as a noncompetitive inhibitor of IMPDH. | merimepodib | IMPDH | 1 | 1 | 1 |
| EMA/PMA treatments significantly decreased amplifiable hg/cont and significantly increased the number of PVDA positive probes and their signal intensity compared to untreated spiked lung samples. | EMA | hg/cont | 0 | 1 | 0 |
| EMA/PMA treatments significantly decreased amplifiable hg/cont and significantly increased the number of PVDA positive probes and their signal intensity compared to untreated spiked lung samples. | PMA | hg/cont | 0 | 1 | 0 |
| Thus, EMA/PMA treatments offer a new approach to lower the amplifiable hg/cont in clinical samples and increase the success of PVDA and HTS to identify viruses. | EMA | hg/cont | 0 | 1 | 0 |

| | | | | | |
|---|---|---|---|---|---|
| A role for glycosylation in cell-surface tetherin expression is supported by tunicamycin treatment, which inhibits the first step of N-linked glycosylation and impairs both cell-surface expression and antiviral activity. | tunicamycin | tetherin | 1 | 1 | 1 |
| The effects of cholesterol depletion and PI3K/AKT signaling pathway activation during S. agalactiae-human umbilical vein endothelial cells (HUVEC) interaction were analysed by pre-treatment with methyl-$\beta$-cyclodextrin (M$\beta$ CD) or LY294002 inhibitors, immunofluorescence and immunoblot analysis. | MbCD | PI3K | 1 | 1 | 1 |
| Inhibition of complex-type glycosylation with kifunensine, an inhibitor of the oligosaccharide processing enzyme mannosidase 1, had no effect on either the cell-surface expression or antiviral activity of tetherin. | kifunensine | oligosaccharide processing enzyme mannosidase 1 | 1 | 1 | 1 |
| Conversely, results from cell-based small molecule screening studies have shown that the antibiotic hexachlorophene can down-regulate b-catenin in colon cancer cells. | hexachlorophene | b-catenin | 1 | 1 | 1 |
| The CT of tetherin contains an "STS" sequence that is implicated in ubiquitylation, and a highly-conserved tyrosine-based motif, "YxxY", that is essential for clathrin-dependent endocytosis of tetherin, and activation of nuclear factor-$\kappa$B (NF-$\kappa$B) [16] [17] [18] [19] . | tyrosine | tetherin | 0 | 0 | 1 |
| Here we report that hexachlorophene also counteracts the elevated bcatenin levels in EBV-infected B lymphomas. | hexachlorophene | bcatenin | 1 | 1 | 1 |
| Our results suggest that Siah-1 is targeted by both LMP-1 and hexachlorophene with opposite effects. | hexachlorophene | Siah-1 | 1 | 1 | 1 |

| | | | | | |
|---|---|---|---|---|---|
| The hexachlorophene modulation of Siah-1 and b-catenin is independent of p53 and results in reduced expression of cyclin-D1 and c-Myc (target genes of b-catenin), leading to the growth arrest of B lymphoma cells. | hexachlorophene | Siah-1 | 1 | 1 | 1 |
| Alanine substitution of either Arg-76 or Tyr-94 in the N-terminal domain of IBV N protein led to a significant decrease in its RNA-binding activity and a total loss of the infectivity of the viral RNA to Vero cells. | Alanine | IBV N protein | 0 | 0 | 1 |
| Rat microsomal aldehyde dehydrogenase (msALDH) has no amino-terminal signal sequence, but instead it has a characteristic hydrophobic domain at the carboxyl terminus (Miyauehi, K., R. | carboxyl | Rat microsomal aldehyde dehydrogenase | 0 | 0 | 1 |
| Also, the protease Factor Xa, a target of Ben-HCl abundantly expressed in infected cells, was able to cleave the recombinant and pseudoviral S protein into S1 and S2 subunits, and the cleavage was inhibited by Ben-HCl. | Ben | protease Factor Xa | 0 | 0 | 1 |
| Since the 3a protein forms ion channels, we were interested to see any conformational changes occurring in the Cyot3a upon calcium binding, using fluorescence spectroscopy and circular dichroism. | calcium | 3a protein | 1 | 1 | 1 |
| These studies clearly indicate a significant change in the conformation of the Cyto3a protein after binding with calcium. | calcium | Cyto3a protein | 1 | 1 | 1 |
| Our results strongly suggest that the cytoplasmic domain of the 3a protein of SARS-CoV binds calcium in vitro, causing a change in protein conformation. | calcium | 3a protein | 1 | 1 | 1 |
| Further, the drug hexamethylene amiloride (HMA), but not amiloride, inhibited in vitro ion channel activity of some synthetic coronavirus E proteins, and also viral replication. | hexamethylene amiloride | synthetic coronavirus E proteins | 1 | 1 | 1 |

| | | | | | |
|---|---|---|---|---|---|
| The high prevalence of variants in the G6PD gene found in this analysis suggests that it may be a significant interaction factor in clinical trials of chloroquine and hydrochloroquine for treatment of COVID-19 in Africans. | chloroquine | G6PD gene | 1 | 1 | 1 |
| In an effort to circumvent resistance to rapamycin -an mTOR inhibitor -we searched for novel rapamycindownstream-targets that may be key players in the response of cancer cells to therapy. | rapamycin | mTOR | 1 | 1 | 1 |
| We found that rapamycin, at nM concentrations, increased phosphorylation of eukaryotic initiation factor (eIF) 2a in rapamycin-sensitive and estrogen-dependent MCF-7 cells, but had only a minimal effect on eIF2a phosphorylation in the rapamycininsensitive triple-negative MDA-MB-231 cells. | rapamycin | eukaryotic initiation factor (eIF) 2a | 1 | 1 | 1 |
| Addition of salubrinal -an inhibitor of eIF2a dephosphorylationdecreased expression of a surface marker associated with capacity for self renewal, increased senescence and induced clonogenic cell death, suggesting that excessive phosphorylation of eIF2a is detrimental to the cells' survival. | salubrinal | eIF2a dephosphorylationdecreased | 1 | 1 | 1 |
| Treating cells with salubrinal enhanced radiation-induced increase in eIF2a phosphorylation and clonogenic death and showed that irradiated cells are more sensitive to increased eIF2a phosphorylation than non-irradiated ones. | salubrinal | eIF2a | 1 | 1 | 1 |
| Similar to salubrinal -the phosphomimetic eIF2a variant -S51D -increased sensitivity to radiation, and both abrogated radiation-induced increase in breast cancer type 1 susceptibility gene, thus implicating enhanced phosphorylation of eIF2a in modulation of DNA repair. | salubrinal | eIF2a variant | 1 | 1 | 1 |

| | | | | | |
|---|---|---|---|---|---|
| In addition to its effect on radiation, salubrinal enhanced eIF2a phosphorylation and clonogenic death in response to the histone deacetylase inhibitor -vorinostat. | salubrinal | eIF2a | 1 | 1 | 1 |
| Finally, the catalytic competitive inhibitor of mTOR -Ku-0063794 -increased phosphorylation of eIF2a demonstrating further the involvement of mTOR activity in modulating eIF2a phosphorylation. | Ku-0063794 | mTOR | 1 | 1 | 1 |
| Moreover, glycyrrhizin treatment still enhanced IFN-γ and reduced IL-4 levels in glycyrrhizin-treated mice. | glycyrrhizin | IFN-g | 1 | 1 | 1 |
| While determining the 5' ends of C. elegans actin mRNAs, we have discovered a 22 nucleotide spliced leader sequence. | nucleotide | C. elegans actin mRNAs | 0 | 0 | 0 |
| The actin mRNA leader sequence is identical to the first 22 nucleotides of a novel 100 nucleotide RNA transcribed adjacent, and in the opposite orientation, to the 5S ribosomal gene. | nucleotides | actin mRNA leader sequence | 0 | 0 | 0 |
| The evidence suggests that the actln mRNA leader sequence is acquired from this novel nucleotide transcript by an intermolecular frans-splicing mechanism. | nucleotide | actln mRNA leader sequence | 0 | 0 | 0 |
| We found that the aspartic acid at position 95, previously believed to be required for binding of PSGs to cells, is not required for PSG1 activity but that the amino acids implicated in the formation of a salt bridge within the N-domain are essential for PSG1 function. | aspartic acid | PSGs | 0 | 0 | 1 |
| In vitro expression of a construct containing the Lb gene fused to a portion of the VP4 and 3D genes demonstrated cis cleavage activity that could be blocked by the thiol protease inhibitor E-64. | E-64 | Lb gene | 1 | 1 | 1 |

| | | | | | |
|---|---|---|---|---|---|
| In contrast, conversion of LC3-I/LC3-II could be significantly inhibited by 4-PBA, an ER stress inhibitor, indicating that ORF3-induced autophagy is dependent on ER stress response. | 4-PBA | LC3-I | 1 | 1 | 1 |
| All the studied molecules could bind to the active site of the SARS-CoV-2 protease (PDB: 6Y84), out of which rutin (a natural compound) has the highest inhibitor efficiency among the 33 molecules studied, followed by ritonavir (control drug), emetine (anti-protozoal), hesperidin (a natural compound), lopinavir (control drug) and indinavir (anti-viral drug). | rutin | SARS-CoV-2 protease | 1 | 1 | 1 |
| Using degenerate PCR primers complementary to the most conserved genome regions of adenoviruses, the complete nucleotide sequence of the penton base gene, and partial nucleotide sequences of the DNA polymerase, hexon, and pVII genes were obtained. | nucleotide | penton base gene | 0 | 0 | 1 |
| Estradiol is connected with CD4+ T cell numbers and increases T-reg cell populations, affecting immune responses to infection. | Estradiol | CD4+ | 1 | 1 | 1 |
| It is known that estradiol exerts a protective effect on endothelial function, activating the generation of nitric oxide (NO) via endothelial nitric oxide synthase. | estradiol | endothelial nitric oxide synthase | 1 | 1 | 1 |

Table 8: Sample sentences from CORD-19 predicted by the proposed model to contain a biochemical relation. The results of the manual validation by a domain expert are reported. Chemical and protein columns denote the entities recognized by BERN; and chemical label and protein label columns report the expert label. The expert label is 1 if the recognition is correct and 0 otherwise. Similarly, the relation label is set to 1 if these entities are biochemically related.