# OpenReview forum: "Vapur: A Search Engine to Find Related Protein - Compound Pairs in COVID-19 Literature"
_EMNLP/2020/Workshop/NLP-COVID — NLP-COVID19-EMNLP Poster_

### Official Review · AnonReviewer2 · 2020-09-25
**The paper presents an interesting work, but it is difficult to read. The paper would benefit from revision on the order the information is presented and removing some reptations.**

**Rating:** 7
**Confidence:** 3

**Review:**

Comments:
Introduction: The last paragraph staring with “in order to”, contains technical detail of the work. It would have been better to highlight the contribution and importance of the work and leave the technical description to the next sections.

Figure 1 show the overall architecture of the work but the main component which is CORD-19 is not explicitly included.

For this sentence “Vapur indexes only these abstracts, but it is able to return the linked full-paper,
as well.” What does it mean? As a reviewer I had to decipher it to mean this “ Vapur indexes into these abstracts, and is also able to return the linked full-paper.” Is that correct? (Only and as well do not go together).

 Section 3.3 there seem to be no need to reference BioBERT again.

Where Chemprot dataset is introduced include the number of records (publications).

Table 1 would benefit from having a total row.

Correct: “In order too promote”.

As a reader I found myself going back and forth between sections to try to find how the components are introduced and what the contributions are. Both Results and Discussion and Conclusion sections have Vapur explained in them. I would suggest describing Vapur only in the Conclusion section.

---

> ### Author Response · Authors · 2020-09-27
> **Response**
>
> Thank you very much for your positive assessment of Vapur and the detailed feedback. We appreciate all of your comments will apply them on the manuscript.
>
> **...For this sentence “Vapur indexes only these abstracts, but it is able to return the linked full-paper, as well.” What does it mean?...**: Thank you very much for suggesting the clarification of this sentence. You are correct in your interpretation of Vapur’s indexing approach. We will clarify this part in the manuscript as well. Thanks for pointing this out!
>
> **Writing style issues:** Thank you very much for pointing out the issues with writing style and following the EMNLP reviewer guidelines for your assessment!

---

### Official Review · AnonReviewer1 · 2020-09-25
**[revised -- lean accept assuming revisions] System demo of COVID-19 search tool; lacks proper system-level evaluation with a biomedical expert**

**Rating:** 6
**Confidence:** 4

**Review:**

Per author response, I've revised my review from 4 to 6 contingent on authors including the additional evaluation & clarifications they've described in the thread below.

============

Vapur is a tool for finding related protein-chemical pairs. The idea is to (i) extract & normalize entities using BERN, (ii) identify relations using a ChemProt-supervised model, (iii) index papers based on related entity pairs.  Search results organized by relationships with other potential drug targets and entities of interest.  The authors evaluate the system's predictions in a small manual study.

Since the methods used to build the system are applications of existing methods, and the key innovation is in the interface, I recommend restructuring this paper like a demo paper.
1) Make Figure 3 --> Figure 1.
2) Spend more time walking through how a user should use Vapur for search/exploration/discovery.  Taking us through a compelling example from the perspective of a real biomedical researcher user would be nice  (the current example in Figure 3 caption is a bit lacking in motivation)
3) Add to Related Work a review of other similar relation-oriented COVID19 systems.   Some examples include KDCovid (http://kdcovid.nl/about.html), EVIDENCEMINER (https://www.aclweb.org/anthology/2020.acl-demos.8/), and SemViz (https://www.semviz.org/), which all surface relational information between bio-entities for COVID-19.  I believe Vapur is sufficiently different from theirs because of the use of a relation-based inverted index, but I recommend the authors try to articulate this clearly earlier in their paper to help distinguish their work.
4) Move the details about the Methods (sec 3) to after the reader has familiarized with the system.

Other writing feedback:
- Table 6 is not discussed in the text.
- Please include the Appendix from the GitHub containing domain-expert-checked instances in the final version of this paper.  Don't make us go look for it.

Questions:
- Why did you decide to use 1/0 binary classification of relation instead of predicting the class labels?  Were the relation class labels not useful for your particular user interface? This would be very interesting to discuss

Evaluation feedback:
- It's good you performed an evaluation for the relation extraction module.  When checking the Appendix, I realized the evaluation was on the binary level (i.e. does this express a relation or not?).  While this is fine to identify that errors are coming from entity normalization, I think your system could benefit from an error analysis on the sentences itself (i.e. what types of sentences does the system tend to fail on?  maybe split this out by the existing relation types in ChemProt?)

- I'm disappointed to not see a human evaluation of the system in front of (at least one) real user.  When we build such tools, it's important to get feedback from real biomedical researchers (e.g. given they issue K queries, do they find the results sensible/helpful?)  It doesn't have to be extensive (given this is a workshop submission), but some indication of having verified that this tool is actually something biomedical researchers would find helpful would give readers more confidence that system designs like Vapur are a good idea.

---

> ### Author Response · Authors · 2020-09-27
> **Response I**
>
> Thank you for the detailed feedback you have provided for our paper. Thank you also for pointing out the novelty of the interface. In the revised version, we describe Vapur and its distinguishing properties earlier in the text and add an attractive use case scenario from the perspective of a domain expert as suggested in point 2. We will further emphasize our claims by re-organizing the text based on your comments, adding the suggested references to the manuscript, and extending our evaluation.
>
> **...Spend more time walking through how a user should use Vapur for search/exploration/discovery...**: Noted and added.
>
> **...Add to Related Work a review of other similar relation-oriented COVID19 systems...**: We thank you for this comment. As you pointed out, the main novelty of Vapur is that it uses a relation-based inverted index and it also explores the inherent graph structure for similar entity retrieval. We will extend the Related Work section as suggested and clarify the novelty of the proposed system.
>
> **...Move the details about the Methods... Table 6 is not discussed in the text..:** All noted and added.
>
> **...include the Appendix from the GitHub...:** We will include the annotations from the online appendix in the manuscript for easier reading.
>
> **Why did you decide to use 1/0 binary classification of relation instead of predicting the class labels?**: Even though Vapur uses a ChemProt-supervised relation extraction model, its main goal is to identify biochemically related chemical - protein pairs, independent of the relation type. By formulating the problem as a binary classification (i.e. is there any type of biochemical relation or not), Vapur can potentially identify the biochemically related molecules in CORD-19, whose relation type is not included in ChemProt and retrieve a larger number of related documents. Plus, binarization of the classes creates a simpler formulation than multi-class classification which can improve the overall model performance.
>
> **Evaluation feedback:**
>
> **...system could benefit from an error analysis on the sentences itself...:** We agree that the current evaluation can be enhanced. We already discussed the potential sources of error and plan to detail our discussion with more domain expert insights. We believe that this would enlighten the limitations of our model.
>
> **...disappointed to not see a human evaluation of the system in front of (at least one) real user...**: To evaluate the overall performance of Vapur, we already selected 41 sentences and evaluated the performance of BERN and binary relation extraction in CORD-19 abstracts from the perspective of a domain expert. We plan to extend the existing scheme and currently collaborating with domain experts to this purpose. We asked them to use and rate Vapur in terms of different aspects and will discuss their assessments in the paper.
>
> Thanks again for all your insightful comments!

---

> > ### Comment · AnonReviewer1 · 2020-09-30
> > **Requesting more info from authors**
> >
> > Thanks to the authors for responding!  It seems the authors have made changes to the manuscript based on the feedback; and if this is the case, I would certainly be happy to revise my scores to be higher.
> >
> > Would it be possible for the authors to share more specifics about what they're changing in the paper?  For example, the authors said that they've added something about an attractive use case about how a domain expert would use the tool.  Would it be possible to respond here with what that looks like?
> >
> > Regarding the 1/0 binary classification, I think the point of potentially identifying relations whose relation types aren't included in ChemProt is a very valid point.  I encourage the authors to include this in the manuscript as justification for the 1/0 decision.  Ideally, if there were some analysis of how often this actually occurs, I think this would also strengthen the submission, but if it doesn't exist, that's fine too for a workshop submission.
> >
> > *We asked them to use and rate Vapur in terms of different aspects and will discuss their assessments in the paper.*
> >
> > Regarding this, are you able to provide some more details here?  It looks like Reviewer 4 also had a question around online evaluation, so it's not just me who's concerned about evaluation in a more realistic setting (not just evaluation of the relation extractor).  Seeing some of this discussed here would give me more confidence to raise my review score.
> >
> > Thanks!

---

> > > ### Author Response · Authors · 2020-10-01
> > > **Providing more info**
> > >
> > > We sincerely thank you for your interest in our work and commenting second time! We happily share more details on our plan for your consideration.
> > >
> > > ...**what an attractive use case looks like**...
> > > We consulted with a medicinal chemistry researcher working on COVID-19 vaccine development to design a compelling and realistic use case. Here is the scenario that we will include in the camera-ready version, where the placeholders D1 and D2 will be replaced by the corresponding real entities: The researcher searches for a drug (D1) that can be a potential treatment for COVID-19 on Vapur. She finds the related papers and biochemicals to the drug and also sees another potentially useful drug (D2) among the listed similar entities. She uses the provided links to access the original publications of the results and also decides to extend her research on D2.
> > >
> > > Figure 3 will also be updated accordingly in the camera-ready version.
> > >
> > > ...**potentially identifying relations whose relation types aren't included in ChemProt is a very valid point.... analysis of how often this actually occurs ... would also strengthen the submission...**
> > > We will discuss our motivation behind binary classification in the camera-ready version of the manuscript. We will also try to identify sample biochemical relations annotated by Vapur that are not included in ChemProt and discuss the insights of the domain expert in our team to present a qualitative evaluation.
> > >
> > > ...**are you able to provide some more details [on online evaluation] here?**...
> > > We also agree that an online evaluation would strengthen our work and thank the reviewers for pointing that out! We are currently collaborating with 5 chemists/biologists to rate Vapur. We ask them to come up with 5 biochemical queries that can be used for COVID-19 research and search the results on Vapur. We also ask them to answer the following questions per query:
> > >
> > > - Is the first displayed result a biochemically related entity to your query? (Yes/No)
> > >
> > > - How many of the top 5 displayed results are biochemically related entities to your query? (0, 1, …, 5)
> > >
> > > - How many of the displayed similar entities to your query do you find useful? (0, 1, ..., 5)
> > >
> > > - Do you find the displayed sentence that contains your query useful? (Yes/No)
> > >
> > > We also ask them to rate the overall usefulness of Vapur with the following questions:
> > >
> > > - How would you rate the ease of use of Vapur? (1, 2, ..., 5)
> > > - Do you find Vapur useful for future research on COVID-19? (Yes/No)*
> > >
> > > We thank you again for spending more time on Vapur and being open to our comments!

---

### Official Review · AnonReviewer4 · 2020-09-29
**Useful search engine for biomedical entities COVID-19 related, but needs more work on the evaluation**

**Rating:** 5
**Confidence:** 4

**Review:**

Dear authors,

This paper is about the development of a search engine for Proteins-compounds extracted from CORD-19 dataset.
The pipeline of the work consists in extracting the entities from the abstracts using BERN and normalizing these entities. Then the system uses the BioBERT, previously tuned using the ChemProt dataset, for relation extraction between the entities (e1 for chemicals and e2 for proteins). Finally, they create an inverted index.

I think this is a valuable work, well written, however a little confused in its structure.

The introduction provides a good insight of the work, but I was expecting more information here.
The related work section needs more work. This section talks about NER, BERN, but it does not present RW related to search engines, thus, when the authors state that this is the first search engine that “uses relation extraction to construct an inverted index of related biochemical entities”, I thought that I need more info about search engines provided in the RW.

In Methods, the paper presents very well the datasets used, I have just some issues:
- Why do you consider CPR:0 as 1? Is there a certain relation there but ChemProt doesn't know what is it?
- Here the authors introduce new concepts, such as GENIA, and finally say what is BERN (a neural NER architecture with an integrated normalizer), but I think this should be said previously in the paper.
- I didn’t quite understand what was used for the normalization

About Vapur, it seems a useful search engine, but there is no evaluation here.
Are you planning some online evaluation?

In Results and Discussion the authors only evaluate BERT vs BioBERT performance, which seems a bit limiting and expected that BioBERT performs better in a Biomedical related dataset.

Further questions:
- In vapur there is the display of similar entities, for example, similar genes as in Figure 3. I didn’t understand how the similarity is calculated here.
- Are you considering the use of the full text? Some precious entities may be enclosured there.

---

> ### Author Response · Authors · 2020-09-29
> **Response III**
>
> **Response 3:**
>
> We thank you for your valuable comments on Vapur. We restructure the manuscript in the revised version and address your comments.
>
> ...**a little confused in its structure.**...
>
> Thanks for your comment. We restructured the manuscript in its revised version for easier reading.
>
> ...**The related work section needs more work**...
>
> We agree with your opinion on the related works and extended our references with suggested works in the reviews. We also clarify our novelty in the revised version of the manuscript.
>
> ...**Why do you consider CPR:0 as 1? Is there a certain relation there but ChemProt doesn't know what is it?**...
>
> Unfortunately. ChemProt guidelines do not define CPR:0 and we did not provide a description as well in order to avoid any confusion. We set the corresponding label to 1 since the chemical - protein pair is annotated (indicating a relation information in the context) and its label is different from “NOT”. We also note that only 3 pairs are part of CPR:0 and we view their impact on our relation extraction model as negligible.
>
> ...**Here the authors introduce new concepts, such as GENIA, and finally say what is BERN (a neural NER architecture with an integrated normalizer), but I think this should be said previously in the paper.**...
>
> BERN and GENIA are external tools in our pipeline that we could replace with others in the future and thus we aim to keep the focus on the overall structure and usefulness of Vapur. Therefore, we skip GENIA, a sentence splitter, in the introduction and mention BERN as “*In order to obtain the index, we first identify and normalize the named entities in the documents using a pre-trained model, BERN*”. We thank you for expressing your opinions in this manner and consider to emphasize them more in the introduction.
>
> ...**I didn’t quite understand what was used for the normalization**...
> BERN is a tool to both recognize and normalize the entities in text. We mention the use of BERN for normalization both in introduction, Figure 1, and methods (see Section 3.2)
>
> ....**About Vapur, it seems a useful search engine, but there is no evaluation here. Are you planning some online evaluation?**...
>
> We already evaluated Vapur’s performance from the perspective of a domain expert by handpicking annotated sentences in CORD-19. However, we agree that evaluation can be extended further and currently collaborate with domain experts to add an end-user-oriented evaluation scheme.
>
> ...**In Results and Discussion the authors only evaluate BERT vs BioBERT performance, which seems a bit limiting**...
> We consider experimenting with more language models, such as SciBERT, in our future works.
>
> ...**In vapur there is the display of similar entities, for example, similar genes as in Figure 3. I didn’t understand how the similarity is calculated here.**...
> We use SimRank on the inherent graph structure of our inverted index to identify similar biochemicals and describe our approach in more detail in section 4.
>
> ...**Are you considering the use of the full text? Some precious entities may be enclosured there.**...
> Yes, we plan Vapur as a continuously improving search engine with enhanced models and more data. We used CORD-19 abstracts in the scope of this work to present a proof-of-concept and will include more data sets (such as LitCovid) and full-texts in the future.
>
> We thank you for providing great comments in such a short notice!